



# Soil classification based on spectral and environmental variables

Andre Carnieletto Dotto[1], Jose A. M. Demattê[1], Raphael Viscarra Rossel[2], and Rodnei Rizzo[1]

[1]Department of Soil Science, College of Agriculture Luiz de Queiroz, University of São Paulo, Piracicaba, SP, 13418-900, Brazil

[2]School of Molecular and Life Sciences, Curtin University, Perth, WA, 6102, Australia

**Correspondence:** Jose A. M. Demattê (jamdemat@usp.br)

**Abstract.** Given the large volume of soil data, it is now possible to obtain a soil classification using spectral, climate and terrain attributes. The idea was to develop a soil series system, which intends to discriminate soil types according to several variables. This new system was called Soil-Environmental Classification (SEC). The spectra data was applied to obtain information about the soil and climate and terrain variables to simulate the pedologist knowledge in soil-environment interactions. The most

5  appropriate numbers of classes were achieved by the lowest value of AIC applying the clusters analysis, which was defined with 8 classes. A relationship between the SEC and WRB-FAO classes was found. The SEC facilitated the identification of groups with similar characteristics using not only soil but environmental variables for the distinction of the classes. Finally, the conceptual characteristics of the 8 SEC were described. The development of SEC conducted to incorporate applicable soil data for agricultural management, with less interference of personal/subjective/empirical knowledge (such as traditional taxonomic

10  systems), and more reliable on automation measurements by sensors.

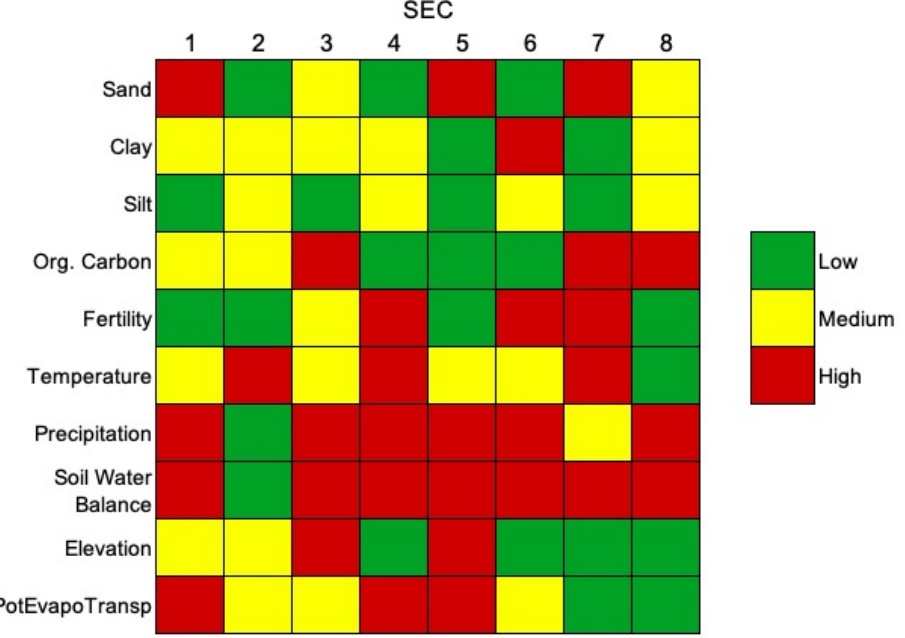

**Keywords.** Soil system, climate, terrain, pedometrics.



# 1 Introduction

The knowledge in soil has been gaining importance since man learned to cultivate the land about ten thousand years ago.
These experiences over the years must be converted into applied knowledge to solve modern issues involving the soil. In
this respect, pedology plays a fundamental role in the understanding of soil formation factors and their spatial distribution.
The pedologist uses his tacit and empirical knowledge to represent the soil by names. The basis of this nomenclature is the
taxonomic classification. Soil classification nomenclature has traditionally been achieved by combining the interpretation of
soil properties, soil-landscape relations along with the support of maps, aerial or satellite images, and with the pedologist
knowledge on soil (Demattê and Terra, 2014). The formative elements of soil classes' nomenclature do not consider climate
or terrain data, which are important factors in the soil formation. Therefore, most of the times there seems to be no coherence
and the comparison is impaired when we try to associate the name of the soil with the landscape. This happens because the
pedologist's knowledge is inherent, acquired with years of learning, it demands time and it is extremely difficult to extract in a
quantitative way. As an alternative, we need to seek sources that aggregate the soil-landscape information into a classification
system.

One source for soil-landscape features can come from remote sensing images. In the last decades, remote sensing is gradually
being applied as a more quantitative technique for soil classes interpretation (Demattê et al., 2004; Mulder et al., 2011; Teng
et al., 2018; Viscarra Rossel et al., 2016). By the digital elevation model is possible to extract several terrain attributes that are
taken into consideration in the soil survey (Florinsky, 2012) by the pedologist. In addition, climate can contribute to a general
understanding of the soil (Brevik et al., 2018) and also, can assist the soil quantification considering the scale of the study.
Climate plays a fundamental role in weathering and soil formation. Terrain attributes present high influence in the soil genesis.
These variables are an essential ally in the search for a better grouping and comprehension of the soil.

Another issue is that traditional soil classification data is turning increasingly challenging to obtain because of the necessity
of the pedologist's knowledge. The complexity and the large number of soil characteristics that should be considered, in order
to classify the soil profile, are another complication. Furthermore, soil classification information is becoming scarce in soil
libraries, since these libraries are greedy for quantitative data. As the traditional approach to obtain the soil classification
data is insufficient, is necessary to introduce new procedures of acquiring soil information in a more measurable way. With
the advent of sensors, the collection and determination of soil data from spectra has become more agile and, with the soil
researches advancement, also more accurate. The application of a quantitative technique is necessary to obtain a new system
to characterize the soils.

How to combine a system that aggregates several soil formation factors without becoming trapped in a taxonomy. From
this question emerged the need for the soil series system. The concept of soil series is a group of soils with homogeneous
characteristics in a system at the lowest possible level. The name of a soil series is the common reference term, used to name
soil map units. The descriptions contain properties that define the soil series and provide a record of soil properties needed to
prepare soil interpretations. Also, a soil series is an area established by similar characteristics of landscape, climate, soils and
therefore, does not involve taxonomy. As pedology got stuck for years in the taxonomy, it had difficulty in creating the soil



series due to the specificity of this new soil denomination, besides the fear of not being easily comprehensible by the user. However, since almost all surveys today are quantitative, including environmental data, the possibility of a soil series system is gain potential. At the moment when homogeneous areas are delimited carrying numerous information about environmental,

terrain and soil, the taxonomic nomenclature of the soil classes will no longer be necessary. In this aspect, the soil spectroscopy is essential. The soil spectrum carries information about soil characteristics such as soil organic matter, minerals, texture, nutrients, water, pH, and heavy metals (Stenberg et al., 2010; Viscarra Rossel and McBratney, 2008). The proximal sensing has presented significant contributions to the soil classification (Viscarra-Rossel et al., 2010) and should play a leading role in the development of the new soil series. However, the spectra data is limited when regarding all information needed in the

soil classification systems. For this reason, environmental data can contribute to supply the inherent pedologist's knowledge in relation to soil-landscape. Besides that, the color, mineralogy, humidity, texture, organic carbon, among others soil properties can be acquired in any part of the world with the same measurement protocol and equipment. Combining this with climatic and terrain data, it is possible to identify areas with homogeneous characteristics.

  The idea is to develop a soil series system that will be called Soil-Environmental Classification (SEC), which intends to

discriminate soil types according to several variables from soil spectra, climate and terrain data. The spectra will be applied to obtain the information about the soil variables, and climate and terrain variables will simulate the pedologist knowledge in soil-environment interactions.

## 2 Material and Methods

### 2.1 Soil data

The soil database consists of 2287 soil profiles from all 5 regions of Brazil. The data was extracted from the Brazilian Soil Spectral Library (Demattê et al. 2019). The database includes profiles of 10 soil classes, classified according to WRB-FAO (IUSS, 2015): Arenosol, Cambisol, Ferralsol, Gleysol, Histosol, Lixisol, Luvisol, Nitisol, Planosol, and Regosol. Each soil profile had three depths, A: 0-20 cm, B: 20-60 cm and C: 60-100 cm. For the statistical analyzes, the spectrum of three depths were averaged to compose a single spectrum per profile. In order to balance the number of samples of each soil class, the

synthetic minority over-sampling technique (SMOTE) algorithm were applied to avoid unbalanced problems in the analyzes (Chawla et al., 2002).

### 2.2 Spectral data

The spectral data were obtained in the Geotechnologies in Soil Science Group (GeoSS), São Paulo, Brazil, using the Fieldspec 3 spectroradiometer (Analytical Spectral Devices - ASD, Boulder, CO). The spectral sensor, which was used to capture light

through a fiber-optic cable, was allocated 8 cm from the sample surface. The sensor scanned an area of approximately 2 cm$^2$, and a light source was provided by two external 50W halogen lamps. These lamps were positioned a distance of 35 cm from the sample (non-collimated rays and a zenithal angle of 30°) with an angle of 90°between them. A Spectralon standard white plate





was scanned every 20 min during calibration. Two replications (one involving a 180°turn of the Petri dish) were obtained for each sample. Each spectrum was averaged from 100 readings over 10 s. The mean values of two replicates were used for each
sample. The spectral data ranged from the visible to near infrared (350 – 2500 nm). The Savitzky-Golay derivative (Savitzky and Golay, 1964) was applied in the spectra with following configuration (polynomial order of 2 and window size of 15). Since the spectrum is highly collinear, we kept only the wavelength in every 10 nm, resulting in 213 wavelengths for the analysis. The soil color variables, which comprises Hue angle (Ha), Value (v) and Chroma (c), were derived from the spectrum.

We applied the principal component analysis (PCA) in the spectral data to select the scores of principal components (PC)
and applied them in the modeling. The PC eigenvectors were utilized to indicate the wavelengths of highest contribution in the PCA. The data was not standardized because all of the wavelengths are in the same units and the differences in variation between them are inherently important. The number of PCs applied in the modeling were selected in order to capture high percentage of the variance explained and the maximal spectral details as possible, since the spectral data present absorption points in different areas of the spectral curve and with distinct intensities.

### 2.3 Climatic and terrain variables

The climatic and terrain variables, applied in the modeling, were extracted from different sources in order to represent the environmental variability. The climatic variables were the Potential EvapoTranspiration (PotEvapoTransp), Soil Water Balance (SWB), Annual Temperature (AnnualTem), and Annual Precipitation (AnnualPre). The terrain variables were Slope, Aspect, Hillshade, Topographic Position Index (TPI), Terrain Ruggedness Index (TRI), Roughness, and Digital Elevation Model
(DEM). The terrain variables were extracted from the DEM (90 m spatial resolution).

### 2.4 Supervised modeling to predict soil classes

In order to evaluate the performance of predicting soil classes, we applied a supervised classification method. Random Forest (RF) was the algorithm selected with 10-fold cross-validation setting. In the first modeling approach, with only the PCs (derived from the spectra) were applied as independent variables (Figure 1). In the second approach, we added the climatic and terrain
variables with the PCs and applied the RF to predict the soil classes. The purpose was to evaluate the improvement when adding climatic and terrain variables to the model. The results were shown by the confusion matrix and the overall accuracy of the model. From the RF model, we were able to obtain the importance of each variable in the classification. The flowchart presents the both classification approaches (1: applying only spectral data, 2: applying spectral, climate and terrain data) and the locations of the soil sites in Brazil (Figure 1).

### 2.5 Unsupervised modeling for the new classification

To derive the classification system, we needed to select the optical number of classes. The unsupervised classification was performed by the k-means clustering analysis. In the first approach, we applied only the spectral data in the k-means clustering. Thereafter, we added climatic and terrain variables with the spectra and performed the k-means clustering again. With this





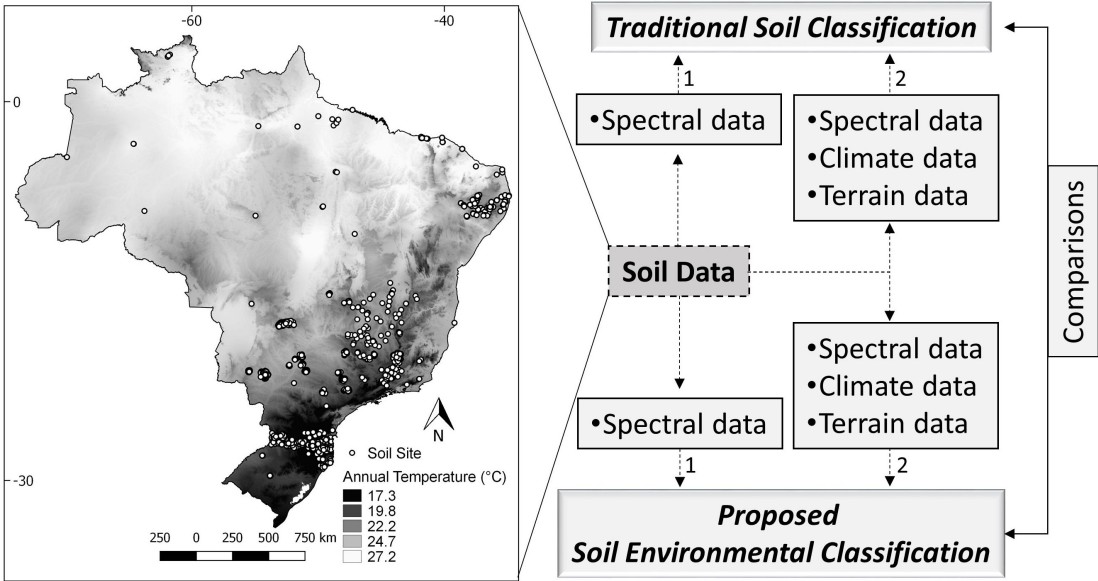

**Figure 1.** Location of soil sites in Brazil and classification approaches 1: Models with only spectral data, 2: models with spectral, climate and terrain data.

procedure, we were able to explore the advantages/disadvantages of adding climate and terrain data to aggregate the groups. To
obtain the optimal number of clusters, the Akaike information criterion (AIC) was used. The lowest value of AIC was assumed
to be the optimal number of clusters, which can represent the most appropriate number of SEC classes. We select the lowest
number by data driven applying 30 times the AIC analysis, then we selected the overall modal value. The AIC was performed
using the values from 1 to 15 clusters.

## 2.6 Soil-Environmental Classification

The number of optimal clusters, performed by the k-means clustering analysis, was referred as SEC classes. The association
between traditional soil classes and SEC was showed by projecting the discriminant coordinates. This procedure allowed to
identify the homogeneity of the classes and additionally, the proximity of the classes and the relation between them. The
correlation between soil classification and SEC classes was arranged in a table. The characterization of each SEC class was
performed by evaluating the relationship between categorical variables, including soil, climate and terrain variables.



## 3 Results

### 3.1 Extracting the principal components of the spectral data

The discrimination by PCA revealed that the first 10 PCs accounted for 94.5% of the variance explained (Figure A1, supplementary data). In order to capture the maximum variation of spectral data, the first 10 PCs were considered as the spectral information to predict the traditional soil classification and to develop the SEC. Vasques et al. (2014) applied 20 PCs to derive the classification models. The eigenvectors of PC1 to PC10 represent the important spectral features and the contributions of the absorbance at individual wavelengths (Figure A2, supplementary data). According to Viscarra Rossel and Webster (2011), the functional groups of minerals and organic components that were most useful in the discrimination of soil classes were those that are related to iron oxides (430, 495 and 570 nm) O – H in minerals (1420 nm), H – O – H in smectite (1900 nm), organics and clay minerals (2150 nm), and Al – OH clay minerals (2250 nm). The wavelengths for these absorption are approximate. For Bishop et al. (2008), the reason they may shift from the expected wavelengths is because real molecules do not behave totally harmonically when they vibrate and also for the reason that the differences in measurement conditions and instrumentation.

### 3.2 Predicting traditional soil classes

The performance of RF model showed an overall accuracy of 83% using only spectral data (Table A1, supplementary data). The confusion matrix and accuracy deriving from the RF analysis applying spectral, climatic and terrain variables are shown in Table 1. The overall accuracy for this classification model by RF was superior reaching 88%. The values in the matrix are the number of samples in each class allocated by the RF model. Three soil classes had an improvement of 10% or more in the prediction when climatic and terrain variables were added to the model. The overall accuracy of Cambisol went from 73.8% to 83.8%, Gleysol from 84.2% to 94.3%, and Ferralsol, which presented the largest improvement, from 56.3% to 72.5%. The RF model with spectral, climatic and terrain data was able to assign the correct soil class in a very prominent accuracies for Histosol, Luvisol, Planosol, Nitisol, and Gleysol reaching values higher than 94.3%. The Ferralsol was the most misclassified class, where the accuracy reached only 72.5% and, consequently 27.8% were re-allocated to other classes but mostly in Arenosol (14%), Lixisol (7%) and Regosol (3%) (Table 1). Regosol showed a class accuracy of 72.9%, and most of its misclassification were re-allocated in Cambisol (13%) and Ferralsol (7%). Cambisol presented relatively moderate class accuracy (83.8%) with most of errors re-allocated in Regosol (6%). Both Cambisol and Regosol classes presented similarities. Regosols comprise soils in unconsolidated deposits that hardly show signs of pedogenesis with no B horizon and Cambisols present beginning of soil formation with weak horizon differentiation. As for the Arenosols (class accuracy of 76%), the misclassification was predominantly observed with Ferralsol, Lixisol and Planosol. The Lixisol (class accuracy of 79.9%) misclassification was also occurred with Ferralsol (9%) and Arenosol (7%), indicating that these three classes present common soil properties. The Arenosols are soils with little or no profile differentiation with texture class of loamy sand or coarser. The majority of Ferralsols in the current data set contained high sand content. The same occurred with Lixisols. These two soil classes presented the sandy characteristic because they are predominantly derived from sandstone rocks. This was the reason




**Table 1.** Confusion matrix and accuracy of soil classification model using spectral, climatic and terrain data.

|  | Arenosol | Cambisol | Ferralsol | Gleysol | Histosol | Lixisol | Luvisol | Nitisol | Planosol | Regosol |
|---|---|---|---|---|---|---|---|---|---|---|
| **Arenosol** | 180 | 1 | 31 | 0 | 0 | 15 | 0 | 0 | 0 | 1 |
| **Cambisol** | 0 | 192 | 7 | 9 | 0 | 3 | 0 | 0 | 0 | 30 |
| **Ferralsol** | 27 | 6 | 166 | 0 | 0 | 21 | 0 | 1 | 0 | 16 |
| **Gleysol** | 0 | 4 | 1 | 215 | 0 | 1 | 0 | 0 | 2 | 6 |
| **Histosol** | 0 | 1 | 0 | 1 | 229 | 0 | 0 | 0 | 0 | 2 |
| **Lixisol** | 13 | 4 | 16 | 2 | 0 | 183 | 0 | 1 | 1 | 3 |
| **Luvisol** | 0 | 1 | 0 | 0 | 0 | 3 | 228 | 0 | 0 | 0 |
| **Nitisol** | 0 | 4 | 7 | 0 | 0 | 2 | 0 | 226 | 0 | 0 |
| **Planosol** | 9 | 2 | 0 | 1 | 0 | 1 | 0 | 0 | 223 | 4 |
| **Regosol** | 0 | 14 | 1 | 0 | 0 | 0 | 0 | 0 | 3 | 167 |
| **Total number of profiles** | 229 | 229 | 229 | 228 | 229 | 229 | 228 | 228 | 229 | 229 |
| **Class Accuracy (%)** | 78.6 | 83.8 | 72.5 | 94.3 | 100 | 79.9 | 100 | 99.1 | 97.4 | 72.9 |
| **Overall Accuracy (%)** | 88 | | | | | | | | | |

why these three classes were not well distinguished by the modeling. Overall, not all misclassifications are negative and some classes are very similar in properties and use, while other classes are radically different.

### 3.3 Variable importance from the soil classification model

The variables importance derive from the RF model were represented in Figure 2. The variable importance for the spectral data is represented by the 10 PCs. PC1 presented an importance by more than 50% to discriminate almost all soil classes, with the exception of Cambisol. PC1 showed significant contribution to distinguish soils with absorption effect on visible region (380 to 740 nm), where the characteristics of iron oxides are present (Figure A2, supplementary data). The PC1 also exhibited important bands related to hydroxyl bonds (1420 and 1900 nm) and with organics and clay minerals peaks (2150 and 2250

nm). The remaining PCs showed important bands in the same features but with variation in intensity. Ferralsols and Nitisols are associated with iron oxides in the visible region of the spectrum, which the PC1 showed high contribution. Planosols contain high clay in the subsurface horizon, which indicates the presence of clay minerals. Histosols are rich in organic minerals, which are presented in the PC1. These soil classes were the ones with the higher variable importance considering the spectral data (PC1 had 47% of variance explained). As the variance explained in the PCs was dropping, its importance in the classification

was reducing as well.

The variables expressing the color characteristic are Ha, v and c. The color, specifically Ha and v, was important to discriminate Ntitsols. Hillshade, TPI, roughness, aspect and TRI showed relatively low to median importance for all classes, and they were most significant to distinguish the Planosols. As this class present impermeable subsoil with significantly more clay in the subsurface horizon and typically located in seasonally waterlogged flat lands, these terrain variables were able to discriminate

it. DEM showed high importance for Lixisols and low for Ferralsols, what indicates that the Ferralsols are located in different



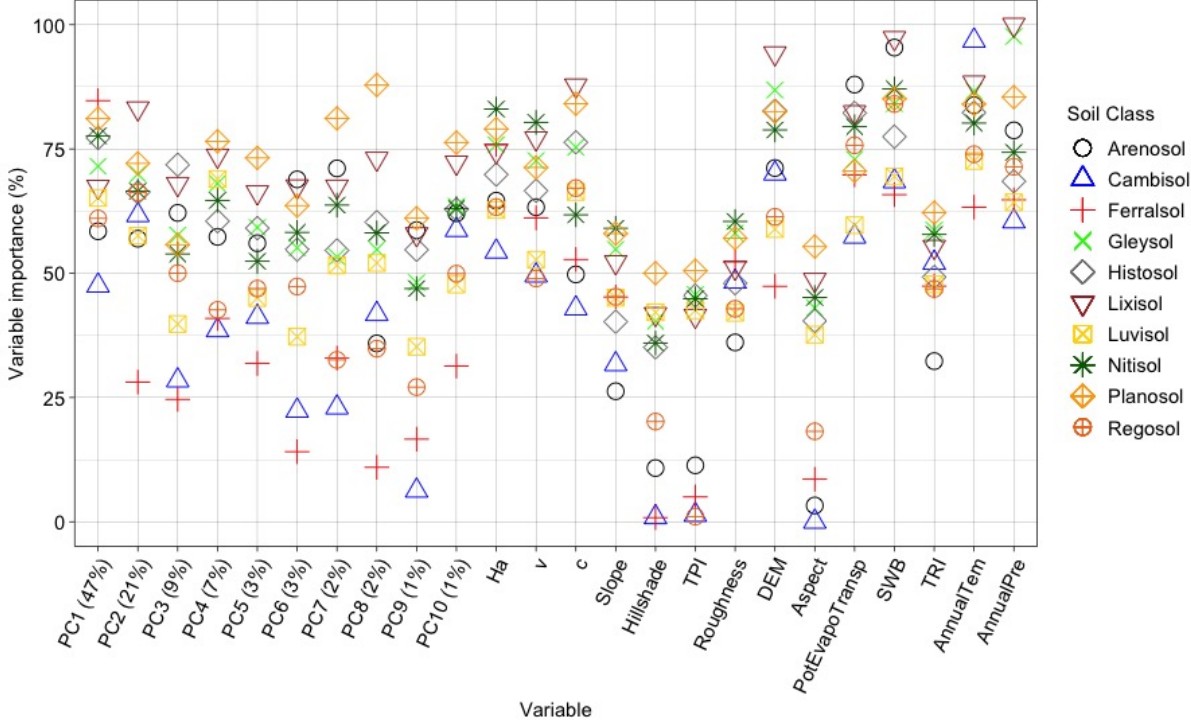

**Figure 2.** Variable importance for each soil class derived from the model using spectral, climatic and terrain data.

sections of the landscape and are not limited to just a certain altitude. The high DEM range effected negatively the importance of this variable to predict Ferralsol. The PotEvapoTransp was most important for Arenosols. Since this soil class present a high sand content, especially in the surface horizon, the PotEvapoTransp is elevated, which contributed to discriminate them. SWB was important variable to discriminate the Lixisols and Arenosols. Lixisols are soils with subsurface accumulation of

low activity clay and high base saturation with moderately drained (because of the argic horizon), they may present a low water retention capacity. SWB refers to the amount of water held in the soil. Because Lixisols are soils that can hold a limited amount of water, there is a risk of percolation in depth or runoff in high precipitations. For the Arenosols, the high content of sand fraction in the entire profile contributed for a high importance of SWB to predict this class. The temperature was important to discriminate the Cambisols, since these soils were located mostly in the South and Southeast regions of Brazil were the average

annual temperature was low. The annual precipitation was important variable for Lixisols and Gleysols. As high precipitation is associated to high soil moisture content and these two soils have an impermeable subsurface horizon condition, superficial water retention and consequently high soil moisture.





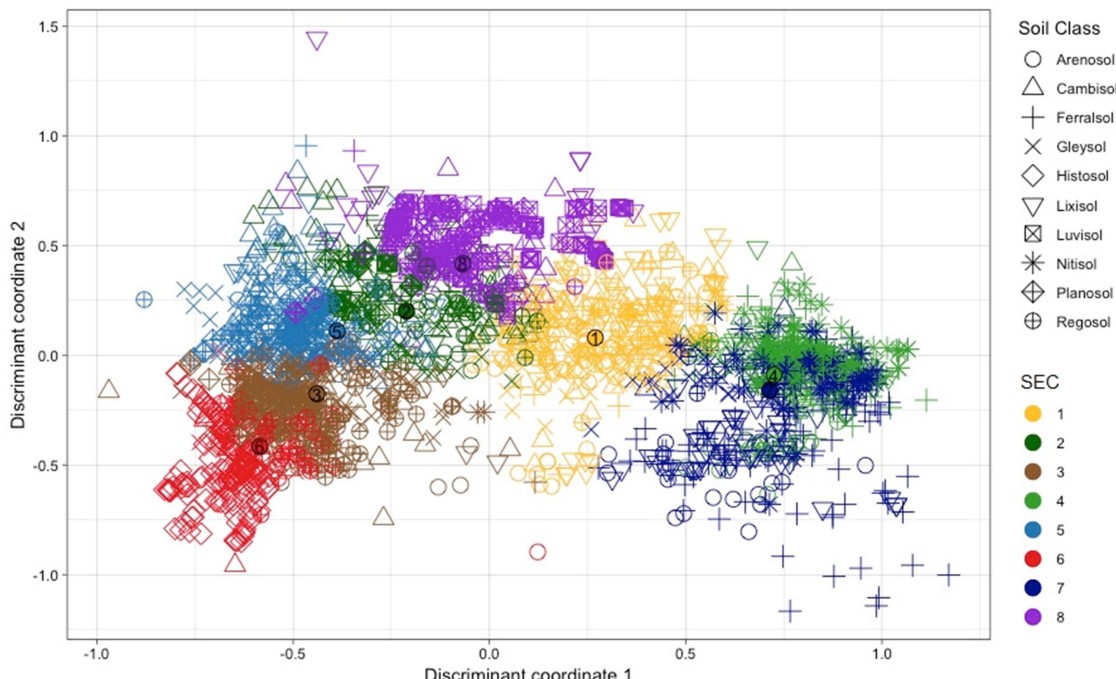

**Figure 3.** Projection of the discriminant coordinates showing the Soil Classification and Soil-Environmental Classification (SEC) applying only spectral data for all samples. The circle with the number represents the center of the SEC.

### 3.4 Developing the Soil-Environmental Classification

The lowest value of AIC was found with 8 clusters (Figure A3, supplementary data), which represent the best spectra catego-
rization. This means that the optimal number of cluster for the current data set is 8. Subsequently, the k-means clustering was performed as the unsupervised classification method applying 8 groups. Firstly, the discriminant coordinates projection, from the clustering analysis using only the spectral data, showed the distribution of the 8 SEC classes (Figure 3). The soil classes located in the left side were soils with less weathering, which is the case of Histosols and Regosols, following by Planosols, Gleysols and Cambisols. In the right side, we can find Ferralsols and Nitisols as more weathered soils. In general, the inter-
mediates weathered soils, such as Luvisols, Lixisols, and Arenosols were located in the center. This tendency proves that the spectral data were able to discriminate soils in different stages of weathering. Arenosols can be considered soils with low level of weathering. However, in the Figure 3, they were close to the classes of Ferralsols and Nitisols. This happened because both Arenosols and Ferralsols have high levels of sand content. For this reason, the spectra curves of both soil classes present soil properties with high similarity.
Because the number of soil classes is greater than the number of SEC classes, it is expected that some soil classes will be allocated in the same SEC class. The association between traditional soil classes with SEC is observed in Table 2. The highest correspondence of SEC 1 was with Arenosols. SEC 2 was associated with Cambisols. SEC 3 presented three soils





**Table 2.** Correlation between Soil Classification and Soil-Environmental Classification (SEC) using only spectral data.

|  |  | Arenosol | Cambisol | Ferralsol | Gleysol | Histosol | Lixisol | Luvisol | Nitisol | Planosol | Regosol |
|---|---|---|---|---|---|---|---|---|---|---|---|
| **SEC** | **1** | 113 | 54 | 47 | 26 | 0 | 63 | 0 | 6 | 0 | 20 |
|  | **2** | 32 | 53 | 12 | 23 | 0 | 22 | 26 | 0 | 23 | 21 |
|  | **3** | 15 | 24 | 1 | 74 | 8 | 5 | 0 | 2 | 105 | 116 |
|  | **4** | 28 | 8 | 40 | 4 | 0 | 42 | 0 | 113 | 0 | 4 |
|  | **5** | 16 | 53 | 4 | 58 | 54 | 21 | 0 | 0 | 71 | 26 |
|  | **6** | 1 | 9 | 0 | 27 | 167 | 2 | 0 | 0 | 12 | 33 |
|  | **7** | 23 | 5 | 123 | 16 | 0 | 47 | 0 | 107 | 0 | 3 |
|  | **8** | 1 | 23 | 2 | 0 | 0 | 27 | 202 | 0 | 18 | 6 |

with high correlation, Regosols, Planosols and Gleysols. The SEC 4 presented high correspondence with Nitisols. SEC 5 had high equivalence with Planosols followed by the Gleysols. SEC 6 was high correlated with Histosols. SEC 7 was correlated with Ferralsols but a great quantity of Nitisols samples was also correlated with this SEC. Lastly, SEC 8 presented the highest correspondence with Luvisols. The Lixisols had no predominant SEC, however presented a high correlation with SEC 1. The SEC 3 and 5 were associated involving three soil classes Regosol, Planosol and Gleysol. The SEC 4 and 7 also presented correlation but in this case with only Ferralsol and Nitisol.

Subsequently, the clustering analysis using spectral, climatic and terrain data were performed. The projection of the discriminant coordinates showed that climate and terrain data reveled that the SEC classes were more gathered (Figure 4), compared to the clustering analysis with only spectral data (Figure 3). The SEC 1 and 3, which correspond mainly to soil classes of Ferralsol, Nitisol and Lixisol, had a more widespread distribution of samples (Figure 4). This arrangement was also observed in the correlation between soil class and SEC using only spectral data (SEC 7, Table 2). Two soils presented association with SEC 2, Luvisols and Planosols (Table 3). SEC 3 showed correlation with Cambisols and Nitisols. The SEC 4 presented only 42 observations, mostly belonging to Gleysols and few to Histosols. These soils were grouped in a specific SEC because they are located in flat lands with DEM close to sea level, with great annual temperature and precipitation, compared to the Gleysols clustered in SEC 6. SEC 5 presented high correspondence with Arenosols and as in the analysis with only the spectra also showed correlation with Ferralsols (SEC 1, Table 2). SEC 6 showed high association with Gleysols. SEC 7 was formed by Histosols and SEC 8 by Regasols. The climate and terrain variables were able to discriminate SEC more properly, but some soils were located far from the center of the class. These soils may have similar properties to other classes but not enough to fit into them.

## 4   Discussion

The Vis-NIR spectroscopy is a technology with the advantages of being faster and cheaper than the traditional soil analysis, with accurate soil classification prediction and enable to acquire the spectra *in situ* (Debaene et al., 2017). Teng et al. (2018) also



**Table 3.** Correlation between Soil Classification and Soil-Environmental Classification (SEC) using spectral, climatic and terrain data.

| | | Arenosol | Cambisol | Ferralsol | Gleysol | Histosol | Lixisol | Luvisol | Nitisol | Planosol | Regosol |
|---|---|---|---|---|---|---|---|---|---|---|---|
| | **1** | 53 | 8 | 146 | 13 | 18 | 160 | 0 | 114 | 9 | 15 |
| | **2** | 21 | 6 | 0 | 0 | 0 | 42 | 228 | 0 | 174 | 2 |
| | **3** | 0 | 121 | 11 | 23 | 34 | 6 | 0 | 109 | 0 | 24 |
| **SEC** | **4** | 0 | 0 | 0 | 30 | 10 | 0 | 0 | 0 | 0 | 2 |
| | **5** | 155 | 12 | 69 | 0 | 0 | 16 | 0 | 5 | 0 | 13 |
| | **6** | 0 | 31 | 2 | 137 | 16 | 0 | 0 | 0 | 0 | 18 |
| | **7** | 0 | 12 | 1 | 24 | 150 | 4 | 0 | 0 | 5 | 10 |
| | **8** | 0 | 39 | 0 | 1 | 1 | 1 | 0 | 0 | 41 | 145 |

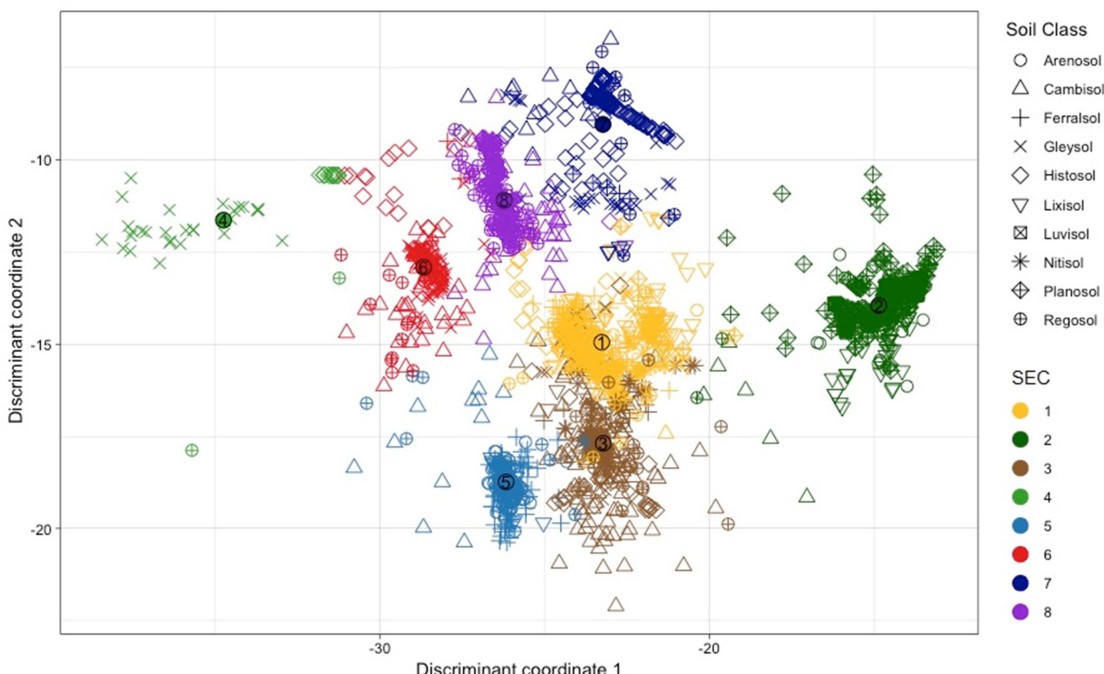

**Figure 4.** Projection of the discriminant coordinates showing the Soil Classification (WRB-FAO) and Soil Environmental Classification (SEC) applying spectral, climatic and terrain data. The circle with the number represents the center of the SEC.

demonstrated the benefit of the technique when updated the Australian Soil Classification with spectroscopic predictions with similar or better correspondence for some classes. In this study, the 10 PCs carried sufficient spectral information to suitably classify the soil as indicated by the overall accuracy (83%) of RF calibration model. Vasques et al. (2014) applied 20 PCs in their study to classify the soil order, and achieved an overall accuracy of 91.6% and 67.4%, for calibration and validation, respectively. The prediction of traditional soil classes applying only the spectral data is considered an excellent prediction



performance. However, when we added climatic and terrain data into the calibration model, this increment of complementary data with the objective of incorporating the Pedologist's impersonal knowledge on environmental, provided an improvement in the prediction of soil classes (overall accuracy of 88%). This increase from 83%, using only spectral data, to 88%, adding climatic and terrain data, may be considered low. However, this result showed that aggregate the soil-landscape information into a classification system improved the prediction of traditional soil classes. Depending on the size of the area and the

characteristic of the study, such addition may not be beneficial, since climatic and terrain data are time-consuming to assemble and are not in a sense practical. Chen et al. (2019) verified the potential of adding auxiliary soil information including color, organic matter and texture for modeling at the soil order levels. They concluded that including such information improved the accuracy of the classification model, although more auxiliary information might be needed for better classification. In general, elevation, slope and relief were the most important terrain predictors in the soil classification for Teng et al. (2018), and

elevation was the most important for hydromorphic soils, which corroborate the findings of the current study where elevation was important variable for Gleysols and Planosols.

Ferralsols, Nitisols and Lixisols presented similarity and were misclassified, and therefore were grouped in the same SEC. However, according to International Soil Classification system of FAO (IUSS, 2015), these two soils are distinct in terms of diagnostic horizons, properties and materials. For instance, Nitisols have nitic horizon, low-activity clay, P fixation, many Fe

oxides, strongly structured, and Ferralsols present ferralic horizon, dominance of kaolinite and iron oxides. These classification differences which distinguish Nitisols from Ferralsols are considered challenging, requiring careful observation by the Pedologist in the field visit. Because both are very similar soils in terms of properties, the spectra tend to have the same shape. The spectral features of Ferralsol and Nitisol showed remarkable similarities in the entire spectral shape and position of absorbance features. Therefore, the spectral distinction between these soils with similar formation processes is reduced. In consequence,

these differences are not perceptible in the spectra indicating that the Vis-NIR spectroscopy was not able to recognize the underlying spectral patterns of each soil class. The main soil properties that influence their soil spectral response is the soil color. In general, resulting from heavy weathering, tropical soils are rich in iron oxides with high contents of hematite, and consequently, showing a red color with lower overall reflectance. Besides, the majority of the soils of this study are developed from Sandstones (sedimentary rock). According to Bellinaso et al. (2010), the distinction between Ferralsols and Nitisols based

on the spectrum is possible but requires an assessment of specific spectral features. However, in the study of Terra et al. (2018), the Vis-NIR spectroscopy methodology could not distinguish Nitisols from Ferralsols. Vasques et al. (2014) found the same misclassification of Nitisols for Ferralsols occurring in 80% of the profiles.

Some classes share many soil properties and even environmental characteristics and are more difficult to distinguish. However, other soil classes are relatively distinctive from the others and, consequently, it is possible to categorize them in a particular

SEC. Soil type differentiation based on the Vis-NIR spectra takes into consideration, predominantly, the soil properties such as color, iron oxides, clay minerals, carbonates and organic matter. According to Viscarra Rossel and Webster (2011), Vis–NIR spectra can be used for the discrimination and identification of soils, when distinguishable mineral and organic characteristics are present in the spectra. The Planosols and Gleysols could be arranged in the same SEC because of their soil properties resemblance. However, they were assembled in distinct SEC. Both soils occurred with seasonally waterlogged areas, poorly



drained, saturated with water for long periods, showing greyish, blueish, reddish, yellowish colors. The main distinction between them is that the Planosols have an abrupt textural difference in the first 100 cm of soil surface. Gleysols have gleyic properties throughout the entire profile. The Histosols were discriminated in a particular SEC. This demonstrates that organic soils are very unique, since they present surface horizons rich in organic matter and B horizons dominated by accumulated organic compounds, characterizing dark colored soils. In the discrimination of Australian soil classes applying Vis-NIR spectra,

Viscarra Rossel and Webster (2011) were also successfully able to differentiate Histosols from the other soils.

     For practical applications (land use and agricultural management), the arrangement of certain classes with similar chemical, physical and/or morphological characteristics is not detrimental, since the decisions about these soils are usually very similar, suffering only minor changes in specific situations (Vasques et al., 2014). Some of the differences between the traditional soil classes are mainly based on specific soil properties and others more on the morphological determination in the field. For

instance, the difference between Ferralsols and Nitisols is minimal and for the new generation of Pedologists this distinction is somewhat tricky. We are not claiming that the role of the Pedologist is not important. On the contrary, there is no way to eliminate it. When is the case of field evaluation to relate the soil-environment formation, the importance of empirical process increases, then when it comes to modeling or digital mapping, this significance diminishes. In terms of agricultural management in natural conditions, Nitisols can provide greater agricultural production, but this may vary for a number of reasons, and

therefore these two classes present practically no management distinctions. For some other classes, such as Cambisols, its classification is intrinsic by the Pedologist to distinguish whether there is or not a presence of sufficient pedogenesis in the subsurface layer to qualify it as Cambic horizon.

     The current soil classification system derived are quite specific to our set of soil classes. We understand the importance of covering the greatest possible number of soil classes. We encourage further research with a larger and diverse types of

soil, possibly in a global level. Despite this, the SEC demonstrated substantial findings regarding the grouping of soils and the utilization of climatic and terrain variables that relates soil-environmental information. As the soil formation is depend on environmental factors, we included climatic and terrain data to simulate the tacit knowledge of the soil-landscape relationship that is taken by Pedologists, who derive traditional soil classification.

     This study sought to develop a classification system using quantitative data. The addition of climate and terrain data was

beneficial and positively collaborated to better distinguish the SEC classes. Moreover, the 8 SEC classes can be individually categorized by observing their soil, climate and terrain properties. The generalised relationship between SEC classes and these properties are shown in Figure 5. The outcomes showed that this classification system proposition could group soil with similar properties. This study can assist the universal soil system, which demand less interference of soil analysis, less personal/subjective data, less need of Pedologists, since their inferences are qualitative, and more use of automated devices,

sensors that can get quantitative information regarding the soil. Lastly, we described the concept and characterization of each SEC. Figure 6 is showing the shapes of each spectral curve for all SEC classes.

     • SEC 1: Soils with high sand, medium clay and low silt contents, medium organic carbon, low fertility, annual temperature around 22°C, high annual precipitation, soil water balance and potential evapotranspiration and located in medium elevation.



• SEC 2: Soils with low sand and medium clay and silt contents, medium organic carbon content, low fertility, annual temperature around 23°C, low annual precipitation and soil water balance, with medium potential evapotranspiration and located in medium elevation.

• SEC 3: Soils with similar sand and clay contents (medium) and low silt content, high organic carbon, medium fertility, annual temperature about 20°C, high annual precipitation and soil water balance, with medium potential evapotranspiration and located in high elevation (in irregular/roughness areas).

• SEC 4: Soils with low sand, medium clay and silt contents, low organic carbon, high fertility, high annual temperature around 26°C and annual precipitation, high soil water balance and potential evapotranspiration and located in low elevation.

• SEC 5: Soils with high sand and low clay and silt contents, low organic carbon content, low fertility, annual temperature around 22°C, high annual precipitation and soil water balance, with high potential evapotranspiration and located in high elevation.

• SEC 6: Soils with low sand, high clay and medium silt contents, low organic carbon content, high fertility, annual temperature around 21°C, high annual precipitation and soil water balance, with medium potential evapotranspiration and located in low elevation areas.

• SEC 7: Soils high sand and low clay and silt contents, high organic carbon content and fertility, high annual temperature around 23°C, medium annual precipitation, high soil water balance, with low potential evapotranspiration and located in low 310 altitudes.

• SEC 8: Soils with relatively balanced sand, silt and clay contents, high organic carbon content, low fertility, low annual temperature (19°C), high annual precipitation and soil water balance, with low potential evapotranspiration and located in low elevation.

## 5 Conclusions

We proposed a soil system that takes into consideration data from spectra, climatic and terrain, and because of these characteristics we called Soil-Environmental Classification (SEC). The SEC was defined with 8 classes according to AIC criteria by clustering analysis. Soil classes like Ferralsols and Nitisols share many soil and environmental characteristics and are difficult to distinguish, however other soil classes, such as Histosols, are relatively distinctive from the others and, consequently, it was possible to categorize them in a particular SEC. This innovative soil system facilitated the identification and grouping of soils 320 with similar characteristics due to the use of environmental variables. The conceptual characteristics of the 8 SEC classes can incorporated more soil information for the agricultural management, with less interference of personal/subjective information, and more reliable on automated measurements by sensors. We believe that this classification system can provide extra information needed for the better understanding and sustainable management of soil. The development of soil systems such SEC can assist in the distinction of soil types and serve as a new soil data source.



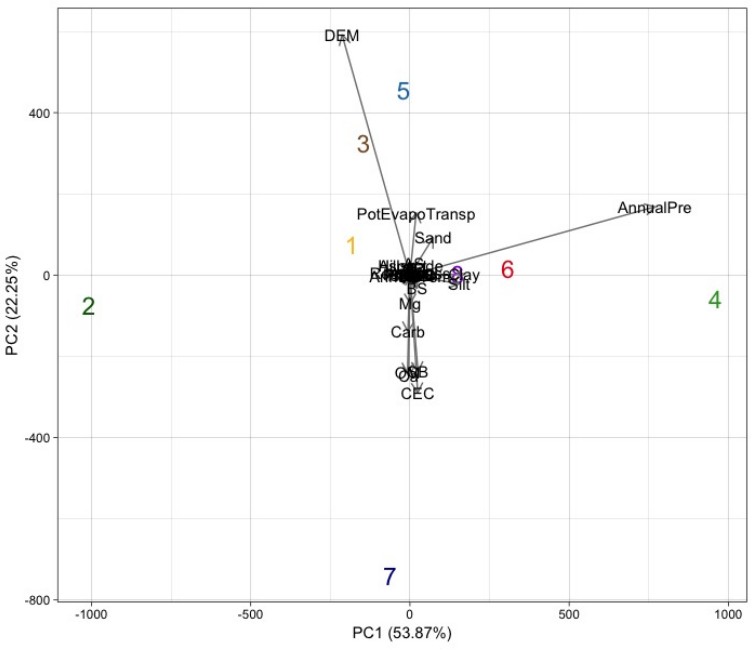

**Figure 5.** Generalised relationship between variable and SEC class.

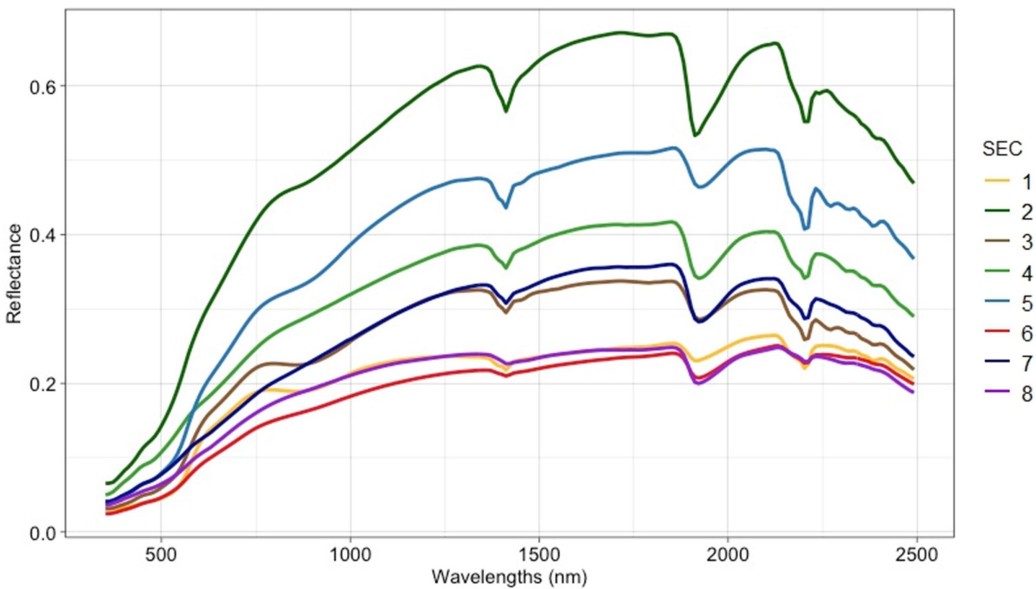

**Figure 6.** Generalised spectral curve of each SEC class.

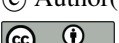



*Acknowledgements.* The first and second authors would like to thank the São Paulo Research Foundation (FAPESP) for the financial support (Projects grant: 2017/03207-6, and 2014/22262-0).

*Author contributions.* A.C.D.: developed the manuscript and performed the analyses. J.A.M.D., R.V.R.: supervised the project, conceived the study and were in charge of overall direction and planning. R.R.: wrote the manuscript, conceived inputs. All authors discussed the results and contributed to the final manuscript.

*Competing interests.* No competing interests are present.



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

**Appendix A: Supplementary data**




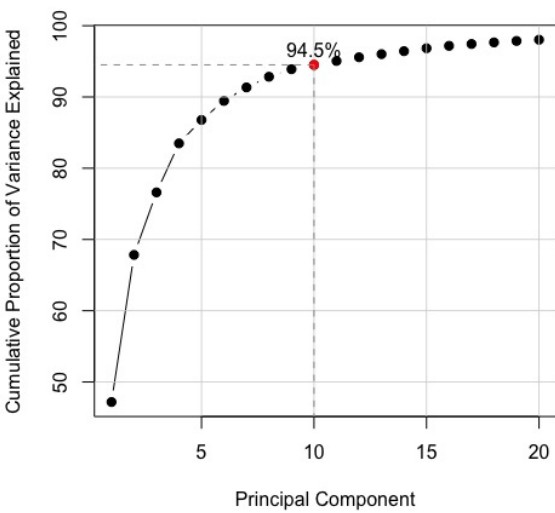

**Figure A1.** Cumulative variance explained for the 10 principal components.

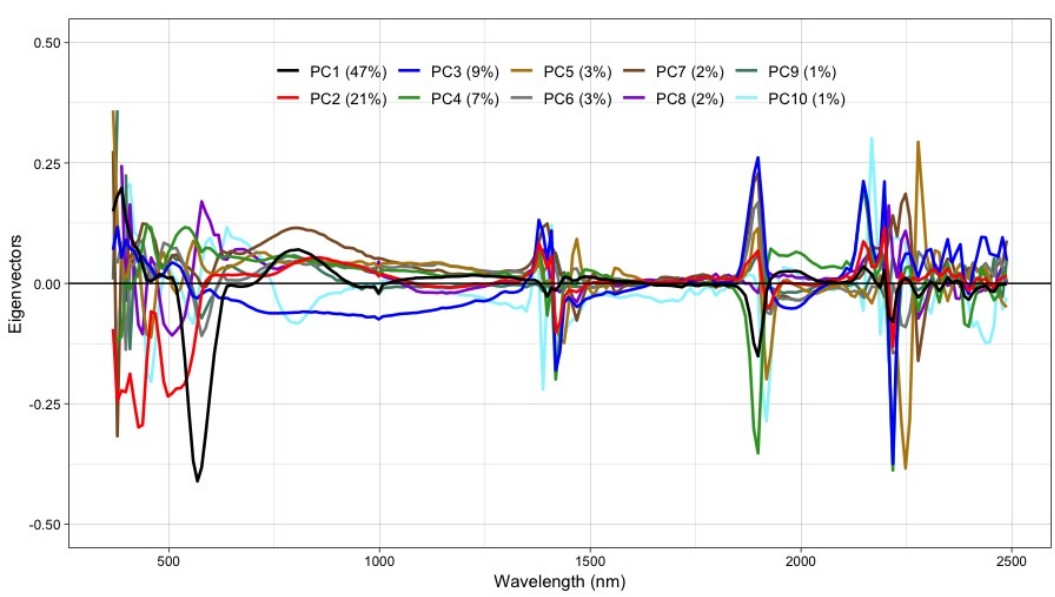

**Figure A2.** The important spectral features and the contributions of individual wavelengths for PC1 to PC10.



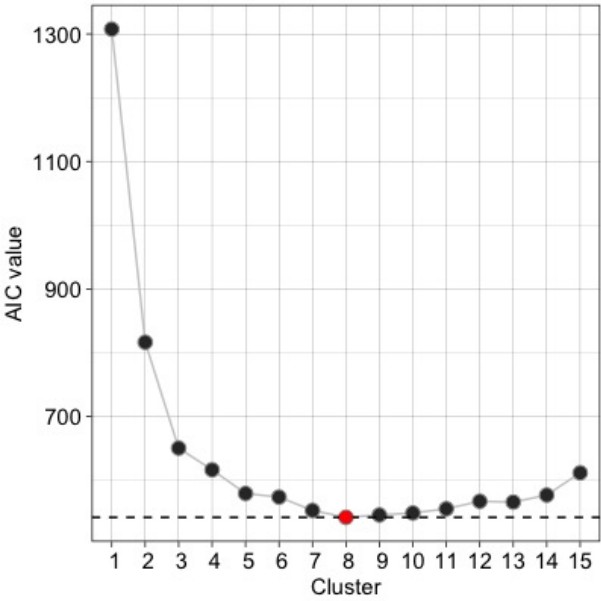

**Figure A3.** The AIC criteria showing that the lowest value was found with 8 clusters.

**Table A1.** Confusion matrix and accuracy of soil classification model using only spectral data.

|  | Arenosol | Cambisol | Ferralsol | Gleysol | Histosol | Lixisol | Luvisol | Nitisol | Planosol | Regosol |
|---|---|---|---|---|---|---|---|---|---|---|
| **Arenosol** | 174 | 4 | 29 | 0 | 0 | 17 | 0 | 0 | 0 | 1 |
| **Cambisol** | 4 | 169 | 8 | 9 | 0 | 5 | 0 | 0 | 0 | 30 |
| **Ferralsol** | 18 | 14 | 129 | 4 | 0 | 21 | 0 | 4 | 0 | 11 |
| **Gleysol** | 5 | 10 | 3 | 192 | 0 | 5 | 0 | 0 | 0 | 10 |
| **Histosol** | 0 | 3 | 0 | 4 | 228 | 1 | 0 | 0 | 0 | 4 |
| **Lixisol** | 20 | 3 | 33 | 1 | 0 | 167 | 0 | 1 | 1 | 5 |
| **Luvisol** | 1 | 0 | 0 | 0 | 0 | 5 | 228 | 0 | 0 | 0 |
| **Nitisol** | 0 | 4 | 26 | 0 | 0 | 3 | 0 | 222 | 0 | 0 |
| **Planosol** | 7 | 4 | 0 | 9 | 1 | 5 | 0 | 0 | 228 | 4 |
| **Regosol** | 0 | 18 | 1 | 9 | 0 | 0 | 0 | 0 | 3 | 164 |
| **Total number of profiles** | 229 | 229 | 229 | 228 | 229 | 229 | 228 | 228 | 229 | 229 |
| **Class Accuracy (%)** | 76.0 | 73.8 | 56.3 | 84.2 | 100 | 72.9 | 100 | 97.4 | 99.6 | 71.6 |
| **Overall Accuracy (%)** | 83.14 | | | | | | | | | |