# Peer review of "Soil classification based on spectral and environmental variables"

_SOIL, 2019_

## Referee Comment (RC1) · Anonymous Referee #1 · 4 Jan 2020

This manuscript presents an idea for a new soil classification system based on spectral, terrain and climate data as a new source of soil information and compares it with traditional soil classification system World Reference Base for Soil Resources (WRB). As this method does not rely on tacit and empirical knowledge of soil scientists but entirely on quantitative measurements, the authors claim it is more robust against subjective evaluation and should be fully transferable across different regions and thus will give coherent information. The authors argue that this approach is different from traditional nomenclature as it takes into account climate and terrain information, whereas the traditional one does not. I do not agree as the traditional nomenclature such as WRB always take into account climate and terrain information when naming soils, e.g. Ferasols occur almost exclusively in humid tropics, because certain climate is a basic

condition for certain soil forming process and soil layers formation. For a good while I was also wondering if the proposed methodology, which is k-means clustering, is an adequate and sufficient method for new soil classes retrieving. The k-means clustering is already firmly adapted in soil science and broadly used for stratification the study area for purposes such as sampling design optimization, or delineating the most diverse classes based on terrain data, etc. In my opinion the methodology presented lacks for deeper quantification of the relations between soil classes and environmental variables. Also, time as one of the key soils forming factors is neglected. Nevertheless, the product presented can be found as a useful extra soil information (treated as environmental co-variate) that can be used within digital soil mapping framework to increase the accuracy of digital soil maps. However, from this point of view, the question remains, whether the classification presented here will do a better job compared to traditional soil taxa (e.g. WRB) or not. One more limitation of the method is that the new classification based partially on soil spectra could not provide quantitative information about soil properties without a proper calibration or linking to known soil taxa. In other words, when applied to unknown area the resulted soil classes will not contain information about soil characteristics. It is a bit unclear how the optimal number of classes was determined. The authors said according to AIC, but there is no equation. Please describe in detail how AIC was computed and provide an equation. Explain why do think the traditional classification system WRB does not take into account climate and terrain information. In summary, the manuscript is written with consistency and clarity, and the overall quality of the presentation and writing is very good. Although I personally do not think that this is the right way that soil science should take further, i.e. to suppress traditional soil scientist knowledge based on experience and replace it with automated procedures, there is certain novelty contained in this study that defends publication in the SOIL journal. Therefore I recommend acceptance after minor revisions.

---

## Referee Comment (RC2) · Anonymous Referee #2 · 6 Jan 2020

It is an interesting manuscript on the exploitation of spectral data from soil profiles for soil classification. The results showed that some classes are well distingushed by spectral features (like Luvisols), while others are hard to distinguish (e.g. Arenools, Lixisols). It was also shown that adding terrain and climatic data improves the classification performance. But there are some issues remained uncleared, which need to be responded by the authors.

- What if just topsoil samples and/or measurements are available, can this classification work?

- The WRB system has 32 reference soil groups, while here just 10 classes were considered. It reflects the classes present in Brazil. Nevertheless, would it work similarly if other classes were also added? Try to discuss this issue at least theoretically.

[Figure]

- On my opinion it would be better to work on the current soil classification system rather than build a new one. Nevertheless, I see the potential of this approach in the possibility to improve the current system, to precise the distinction between classes to make them more easily distinguishable. Then the spectral data could help to identify the classes better.

- I misunderstood the way the spectral were treated (2287 soil with 3 horizons each). First the horizons were arbitrary selected – How can you derive real information about the soil horizons which are mandatory for soil classification? This arises a question: Does spectroscopy is sufficient to classify soils? I am not yet convinced from this study, as a profile has to be well sensed in situ and not from SSL. (see Ben-Dor, Eyal, Daniela Heller, and Alexandra Chudnovsky. "A novel method of classifying soil profiles in the field using optical means."Soil Science Society of America Journal 72, 4 (2008): 1113-1123.

- How the spectral information of each profile was analyzed? The soil orders were dependent values? The spectral were independent? If this soils belong to the SSL of Brazil, why not adding soil attributes such as clay content, organic matter and carbonates to the classification system that can be derived very easily?

- I did not understand figure 6. What exactly is shown here? Average of spectral that represents SEC? How was it done? No variation in each SEC? Needs more clarification. I would be more happy to see spectra profile represented each soil order, which show nice spectral variation from SEC to SEC.

- To be fully convinced, this kind of studies (just spectral information) should be applied on to another SSL from another region.

- Line 106. "optimal" should be instead of "optical".

Line 128. the 1900 nm can be H20 (also at around 1400 nm) from other sources than smectite!

- Line 138. why do you think these soil order provided the "largest improvement" relative to the others? Please explain.

- Line 158. "iron oxide are presented"

- Based on line 228 where it says "may be considered low" – perhaps you should re-title the paper?

- As mentioned color is important . Nonetheless, spectroscopy can depict the color. Please add it to the discussion.

- In several cases it says that incorporating the environmental factors in the analysis is low (line 227) and in others it concluded that it is good (line 285). It makes the reader a bit confused about the authors' opinion: is it good or bad to add environmental factors?

- Line 321. "incorporate" should be instead of "incorporated".

Generally, I would see the manuscript like minor revision.

---

## Author Comment (AC1) · 5 Feb 2020

Referee #1 Referee: For a good while I was also wondering if the proposed methodology, which is k-means clustering, is an adequate and sufficient method for new soil classes retrieving. Answer: K-means clustering is a widely used partitioning method not limited only in soil science but in many other areas. It is widely used as partitioning method to grouping similar sets of data. As every technique, k-means clustering also present its advantages and disadvantages. We selected k-means because its quantization advantages: relatively simple to implement, scales to large data sets, can set the positions of centroids, easily adapts to new examples, and generalizes to clusters of different shapes and sizes. Referee: the classification presented here will do a better job compared to traditional soil taxa (e.g. WRB) or not. Answer: As we have mentioned

in the manuscript, our aim is to provide a new and quantitative approach to define the soil classification based on proximal and remote sensors data. The biggest problem in soil classification is the need for experienced personal (pedologists) to perform the classification. Furthermore, the process requires tacit knowledge from the professional and is relatively subjective. Therefore, the classification process is costly, time demanding and might provide results with large uncertainty. We kindly ask the referee to observe the fact that "doing a better job" is not only related to the reduction in classification uncertainties, but also to define means to perform a fast and reliable classification, which can support soil characterization and consequently soil-related decision processes. We do acknowledge that soil spectroscopy is not capable to distinguish some of the important variables required in conventional classification, e.g. morphological attributes. Some classes share many soil properties and even environmental characteristics and are more difficult to distinguish. However, other soil classes are relatively distinctive, consequently, it is possible to categorize them. Besides that, soil type differentiation based on the Vis-NIR spectra takes into consideration some of the most important attributes, such as soil colour, iron oxides, clay minerals, carbonates and organic matter. The soil spectral classes conducted to incorporate applicable soil data for agricultural management, with less interference of personal/subjective/empirical knowledge, and more reliable on automation measurements by sensors. Referee: Please describe in detail how AIC was computed and provide an equation. Answer: We added more information in the methodology to describe with more detail the AIC analysis. "To determine which number of clusters appears to best describe the data, i.e. the optimal number of clusters, the Akaike information criterion (AIC) was performed. To calculate the AIC, we applied the function kmeansAIC from kmeansstep R package. It calculates the AIC value of a specific k-means cluster and it specified centroids. The AIC was implemented using the values from 1 to 15 clusters. The analysis was performed by data driven 30 times. The overall modal cluster with the lowest AIC value was selected and assumed to be the optimal number of clusters, which can represent the most appropriate number of SSC classes." Referee: Explain why do think the traditional classification

system WRB does not take into account climate and terrain information. Answer: We do not agree that the traditional classification systems does not take into account climate and terrain data, in fact, we have verified exactly the opposite in this manuscript. But we believe that it is difficult to take into account the climate and terrain data in the classification approach. As alternative, we applied remote sensing images as a source of soil-landscape information. By the digital elevation model is possible to extract several terrain attributes that are taken into consideration in the soil surveys, for example. Along with spectral data, we added these climatic and terrain data into the modelling. Adding climatic and terrain data into the calibration model, provided an improvement in the prediction of soil classes. The increment of climatic and terrain data aimed to incorporate the Pedologist's impersonal knowledge on environmental into the quantitative modelling.

---

## Author Comment (AC2) · 5 Feb 2020

Referee #2 Referee: What if just topsoil samples and/or measurements are available, can this classification work? Answer: The new soil spectral classification approach proposed to distinguish soil spectra into 8 classes. If only that is considered when it is plausible to develop a classification with any spectral measurement available. However, in this manuscript we used spectra samples from soil profiles because we wanted; first: evaluate the performance of spectra to predict WRB soil classes, second: classify the spectra into types and compare them with the traditional soil classes. We are focusing in the relationship between WRB classes and the spectral classes and how these spectral types can be a new source for soil management. Therefore, if you want to predict the spectral classes applying only a topsoil spectral sample, the output will

fit in one of the classes, then the understanding and usability of this information it is up to the user. Referee: The WRB system has 32 reference soil groups, while here just 10 classes were considered. It reflects the classes present in Brazil. Nevertheless, would it work similarly if other classes were also added? Try to discuss this issue at least theoretically. Answer: We have mentioned this experimental concern in the discussion. "The current soil classification system derived are quite specific to our set of soil classes. We understand the importance of covering the greatest possible number of soil classes. We encourage further research with a larger and diverse types of soil, possibly in a global level. Despite this, the SSC demonstrated substantial findings regarding the grouping of soils and the utilization of climatic and terrain variables that relates soil-environmental information. As the soil formation is depend on environmental factors, we included climatic and terrain data to simulate the tacit knowledge of the soil-landscape relationship that is taken by Pedologists, who derive traditional soil classification." Referee: On my opinion it would be better to work on the current soil classification system rather than build a new one. Nevertheless, I see the potential of this approach in the possibility to improve the current system, to precise the distinction between classes to make them more easily distinguishable. Then the spectral data could help to identify the classes better. Answer: The idea of this research was to bring innovation to soil systems to facilitate the identification and grouping of soils with similar characteristics applying spectral and environmental variables. New approaches like this can serve as a new soil data source. We believe that this classification system can provide extra information needed for the better understanding and sustainable management of soil. Referee: I misunderstood the way the spectral were treated (2287 soil with 3 horizons each). First the horizons were arbitrary selected – How can you derive real information about the soil horizons which are mandatory for soil classification? This arises a question: Does spectroscopy is sufficient to classify soils? I am not yet convinced from this study, as a profile has to be well sensed in situ and not from SSL. (see Ben-Dor, Eyal, Daniela Heller, and Alexandra Chudnovsky. "A novel method of classifying soil profiles in the field using optical means."Soil Science Society

of America Journal 72, 4 (2008): 1113-1123. Answer: First, it is proved by literature that reflectance spectroscopy is able to classify soils with high accuracy (Bellinaso et al., 2010; Chen et al., 2019; Debaene et al., 2017; Demattê, 2016; Rizzo et al., 2014; Shi et al., 2014; Vasques et al., 2014; Zeng et al., 2016). In our manuscript, we applied a large spectral data set from the Brazilian soil spectral library. In the study mentioned (Ben-Dor et al., 2008), the authors used only 4 profiles to examine and demonstrate the idea. They used different spectral measurement approaches to represent horizons A, B, C in field. The spectral reflectance varies according to the measuring conditions. In the field, illumination can vary a lot as well as soil humidity. This can cause limitations in the spectral scan in situ. For these reasons, most of studies presented the soil spectra were obtained in a standard controlled environmental, which facilitates the homogenization of all spectral samples. This can bring a lot of beneficial aspects to the statistical modelling and reliability. Referee: How the spectral information of each profile was analyzed? The soil orders were dependent values? The spectral were independent? If this soils belong to the SSL of Brazil, why not adding soil attributes such as clay content, organic matter and carbonates to the classification system that can be derived very easily? Answer: The spectra, climatic and terrain data were applied as independent variables. The soil classes were the dependent variables. The soil attributes information was not added in the new classification system because the spectral data already include this information. The spectra can infer various soil attributes and if you use spectra to predict the soil attributes and use these attributes in the modelling is essentially the same approach. Since our objective was to use the spectral data to predict the soil classes and derive a new spectra-based classification system, we employed the soil reflectance and not the soil attributes predicted by them. Besides that, using predicted attributes as inputs in our modelling could increase the error propagation. Therefore, we decided to stablish a direct modelling process, i.e. calibrate prediction models directly from spectral data. Referee: I did not understand figure 6. What exactly is shown here? Average of spectral that represents SEC? How was it done? No variation in each SEC? Needs more clarification. I would be more

happy to see spectra profile represented each soil order, which show nice spectral variation from SEC to SEC. Answer: The figure 6 is showing the shapes of each spectral curve for each class. The spectral curves were averaged after the unsupervised classification discriminate the 8 classes. For the statistical analyses, the spectrum of three depths were averaged to compose a single spectrum per profile. For this reason, the figure 6 is showing the generalised spectral curve of each class. The difference in each spectral curve indicates that soil classes like Ferralsols and Nitisols share many soil and environmental characteristics and are difficult to distinguish, however other soil classes, such as Histosols, are relatively distinctive from the others and, consequently, present distinctive spectral reflectance. Referee: To be fully convinced, this kind of studies (just spectral information) should be applied on to another SSL from another region. Answer: We have mentioned this in the discussion. We understand the importance of covering the greatest possible number of soil classes and other regions. We encourage further research with a larger and diverse types of soil, possibly in a global level. Referee: Line 106. "optimal" should be instead of "optical". Answer: Yes. Referee: Line 128. the 1900 nm can be H20 (also at around 1400 nm) from other sources than smectite! Answer: Yes, we fixed in the manuscript. Referee: Line 138. why do you think these soil order provided the "largest improvement" relative to the others? Please explain. Answer: Because the majority of Ferralsols in the current data set contained high sand content. The same characteristic occurred with Lixisols and Arenosols. These three soil classes presented high sandy content because they are predominantly derived from sandstone rocks. When soil-landscape information was aggregated the into a classification system the prediction not only of Ferralsol but all classes were improved. Ferralsols share many soil properties with other classes. The environmental variables present characteristics that increased the distinction with similar classes like Arenosols and Lixisols. Figure 2 shows these differences between soil classes for the climatic and terrain variables. The elevations, for instance, is one variable that helped to distinguish Ferralsols from other soils. Same with annual temperature. The soil water balance shows a far difference in the distinction of Ferralsols

with Arenosols and Fixisols. Referee: Line 158. "iron oxide are presented" Answer: Yes. Referee: Based on line 228 where it says "may be considered low" – perhaps you should re-title the paper? Answer: We modified this affirmation to become less confusing in the manuscript. The aggregation of soil-landscape information into a classification system improved the prediction of traditional soil classes. Referee: As mentioned color is important . Nonetheless, spectroscopy can depict the color. Please add it to the discussion. Answer: Yes, we added more information in the results: The soil colour is one of the main soil properties that influence soil spectral response. The variables expressing the colour characteristic are Ha, v and c. The colour, specifically Ha and v, was important to discriminate Nitisols, which are heavy weathered tropical soils showing a red colour with lower overall reflectance. And discussion: The main soil properties that influence their soil spectral response is the soil colour, which is an important characteristic used as a criterion in soil type identification (Marques et al., 2019). The colour is usually determined visually in the field by a soil expert. As soil spectral measurements at visible range are related to attributes such as soil organic matter, minerals, texture, nutrients, water, etc, soil colour can be determined using spectroscopic data. Referee: In several cases it says that incorporating the environmental factors in the analysis is low (line 227) and in others it concluded that it is good (line 285). It makes the reader a bit confused about the authors' opinion: is it good or bad to add environmental factors? Answer: The right affirmation is adding environmental variables the prediction of soil classes improves. We changed this phases in the manuscript to help the readers' understanding. Referee: Line 321. "incorporate" should be instead of "incorporated". Answer: Yes.   Reference Bellinaso, H., Demattê, J.A.M., Romeiro, S.A., 2010. Soil Spectral Library and Its Use in Soil Classification. R. Bras. Ci. Solo 34, 861–870. https://doi.org/10.1590/S0100-06832010000300027 Chen, S., Li, S., Ma, W., Ji, W., Xu, D., Shi, Z., Zhang, G., 2019. Rapid determination of soil classes in soil profiles using vis–NIR spectroscopy and multiple objectives mixed support vector classification. Eur. J. Soil Sci. 70, 42–53. https://doi.org/10.1111/EJSS.12715 Debaene, G., Bartmiński, P., Niedźwiecki, J., Miturski, T., 2017. Visible and Near-

[Figure]

Infrared Spectroscopy as a Tool for Soil Classification and Soil Profile Description. Polish J. Soil Sci. 50, 1. https://doi.org/10.17951/pjss.2017.50.1.1 Demattê, J.A.M., 2016. From Profile Morphometrics to Digital Soil Mapping, in: Digital Soil Morphometrics. Springer International Publishing, pp. 383–399. https://doi.org/10.1007/978-3-319-28295-4_24 Marques, K.P., Rizzo, R., Carnieletto Dotto, A., Souza, A.B. e, Mello, F.A., Neto, L.G., Anjos, L.H.C. dos, Demattê, J.A., 2019. How qualitative spectral information can improve soil profile classification? J. Near Infrared Spectrosc. 096703351882196. https://doi.org/10.1177/0967033518821965 Rizzo, R., Demattê, J.A.M., Terra, F. da S., 2014. Using numerical classification of profiles based on Vis-NIR spectra to distinguish soils from the Piracicaba Region, Brazil. Rev. Bras. Ciência do Solo 38, 372–385. https://doi.org/10.1590/S0100-06832014000200002 Shi, Z., Wang, Q., Peng, J., Ji, W., Liu, H., Li, X., Viscarra Rossel, R.A., 2014. Development of a national VNIR soil-spectral library for soil classification and prediction of organic matter concentrations. Sci. China Earth Sci. 57, 1671–1680. https://doi.org/10.1007/s11430-013-4808-x Vasques, G.M., Demattê, J.A.M., Viscarra Rossel, R.A.R.A., Ramírez-López, L., Terra, F.S.S., 2014. Soil classification using visible/near-infrared diffuse reflectance spectra from multiple depths. Geoderma 223–225, 73–78. https://doi.org/10.1016/j.geoderma.2014.01.019 Zeng, R., Zhang, G.-L., Li, D.-C., Rossiter, D.G., Zhao, Y.-G., 2016. How well can VNIR spectroscopy distinguish soil classes? Biosyst. Eng. 152, 117–125. https://doi.org/10.1016/j.biosystemseng.2016.04.019